# ON THE "INDUCTION BIAS" IN SEQUENCE MODELS

## ABSTRACT

Despite the remarkable practical success of transformer-based language models, recent work has raised concerns about their ability to perform state tracking, in particular in out-of-distribution (OOD) generalization, such as length extrapolation. In this work, we shift attention to the in-distribution implications of these limitations. We empirically compare the data efficiency of transformers and recurrent neural networks (RNNs) and find that the amount of training data required by transformers grows much more rapidly with state-space size and sequence length than for RNNs. Furthermore, we analyze the extent to which learned state-tracking mechanisms are shared across different sequence lengths. We show that transformers exhibit negligible or even detrimental weight sharing across lengths, indicating that they learn length-specific solutions in isolation. In contrast, recurrent models exhibit effective amortized learning by sharing weights across lengths. Together, these results demonstrate that state tracking remains a fundamental challenge for transformers, even when training and evaluation distributions match.

## 1 INTRODUCTION

Numerous studies have shown that transformer-based models are fundamentally limited in their ability to perform state tracking (for example, Anil et al. (2022a); Dziri et al. (2024)). This contrasts with recurrent networks, which excel at state tracking. The limitations of transformers have been demonstrated as limitations in out-of-distribution (OOD) generalization, specifically length-generalization: after training models on tasks encoded in sequences of a given range of lengths, they were evaluated on sequences with lengths that were not seen during training.

In this work we perform a detailed and systematic empirical study of the *in-distribution* performance of transformer-based models and contrast these with recurrent models. To this end, we train and evaluate a range of representative models on a range of simple state tracking tasks. Independently varying sequence length and size of the state space in these tasks then allows us to discover regularities in the dependence of generalization error on these parameters.

Step-by-step state updates are a natural inductive bias Hastie et al. (2001) in the context of simple state tracking tasks, as they make it possible to reduce complex multi-step dependencies to a sequence of single-step computations. They also allow a model to share weights across multiple different sequence lengths, as it breaks state updates into single-step, repeatable computations. By analogy to the induction step in a mathematical proof, we shall refer to this kind of inductive bias in this work as "induction bias" (sic). We show that the presence (or respectively absence) of an induction bias, or its relative strength, provides a simple explanation for a wide range of the empirical findings we present. Key take-aways from our study include the following:

- We show that there is a distinct difference between the supervision regimes in which transformers and recurrent networks perform well in-distribution.
- We show that transformers can relatively efficiently learn state tracking tasks in-distribution on one (fixed) sequence length at a time, but generalizing in-distribution over multiple sequence lengths requires significantly more training data.
- We present evidence that, unlike recurrent networks, transformers tend to fail at sharing parameters across sequence lengths and instead learn separate solution mechanisms for different lengths.
- We show that the degree of knowledge transfer across multiple different sequence lengths in the in-distribution setting is highly correlated with the ability of a model to length-generalize.

## 2 METHODOLOGY

**Task:** We consider the task of modular addition, where a model is provided a sequence of $n$ integers $\mathbf{x} = (x_1, x_2, \ldots, x_n)$ with each $x_i$ drawn uniformly at random from $\mathbb{Z}_m = \{0, 1, \ldots, m-1\}$. The objective is to compute the sum of the sequence modulo $m$.

**Length Distributions:** For each generated sample, we first determine the sequence length $n \in \{2, \ldots, L\}$, where $L$ denotes the maximum sequence length. We then sample a sequence $\mathbf{x} \in \mathbb{Z}_m^n$ *without replacement* to ensure that every sample in the dataset is unique. We use three distinct strategies for length selection. *Fixed:* The length is held constant at $n = L$. *Uniform:* Lengths are sampled uniformly at random from the set $\{2, \ldots, L\}$. *Short-to-Long:* Sequences are sampled in ascending order of length, exhausting the available sequences for length $n$ before proceeding to $n + 1$.

**Task Formats:** We consider three task formats that vary in the density and structure of the supervision signal. Let $s_k = (\sum_{i=1}^{k} x_i) \pmod{m}$ denote the $k$-th partial sum of the input sequence. The formats, illustrated in Figure 1, are defined as follows:
*Outcome Supervision:* The model is provided the input sequence $\mathbf{x}$ and is trained to predict only the final sum $s_n$. This format provides no intermediate supervision, requiring the model to discover the latent computational logic of the task on its own during training.
*Chain-of-Thought (CoT):* The model is trained to generate the sequence of intermediate partial sums $(s_1, s_2, \ldots, s_n)$ following the input sequence. This decomposes the task into a sequence of iterative applications of the operator.
*Aligned Chain-of-Thought (ACoT):* The model is tasked to output, for each input token $x_i$, the corresponding partial sum $s_i$. While conceptually similar to the scratchpad, this format provides per-token supervision that is aligned with the input.

**Sample Efficiency:** To quantify the data efficiency of a model under a specific task configuration, we define the *minimal sample size* $N^*$ required to learn the task reliably. Let $\mathcal{D}_N$ denote a training set of cardinality $N$, and let $\mathcal{L}_{\text{val}}(\phi; \mathcal{D}_N)$ denote the validation loss of a model trained on $\mathcal{D}_N$ using hyperparameter configuration $\phi \in \Phi$.

We consider a task successfully learned if the minimum validation loss over the hyperparameter grid falls below a convergence threshold: $\min_{\phi \in \Phi} \mathcal{L}_{\text{val}}(\phi; \mathcal{D}_N) \leq \epsilon$, where $\Phi$ is a predefined grid of learning rates and random seeds, and $\epsilon$ is a convergence threshold. Formally, we define $N^*$ as the smallest training set size satisfying this criterion:

$$N^* = \min \left\{ N \in \mathbb{N} : \min_{\phi \in \Phi} \mathcal{L}_{\text{val}}(\phi; \mathcal{D}_N) \leq \epsilon \right\}. \tag{1}$$

In practice, we estimate $N^*$ by performing a binary search over the training set size. Note that both training and validation samples are drawn from the same underlying data generation process, and therefore $N^*$ reflects the *in-distribution* sample efficiency. In addition to low validation loss, we also consider perfect validation accuracy as an alternative success criterion and find no meaningful difference in the results. For the results in Section 3, we use the latter. See Section C.1 for a comprehensive discussion on data-efficiency results.

**Models:** We compare multi-layer decoder-only transformer architecture Vaswani et al. (2017), with two recurrent alternatives: Long Short-Term Memory (LSTM) (Hochreiter & Schmidhuber, 1997) and dense state-space models (Dense-SSMs) (Fan et al., 2024; Terzić et al., 2025; Ebrahimi & Memisevic, 2025). In our Dense-SSM variant, the state update has no additive term: $h_t = A_{x_t} h_{t-1}$, where the transition matrix $A_{x_t}$ is given by a linear function of the input $x_t$, a property shown to enable state tracking in linear recurrent models. See section appendix A for implementation details.

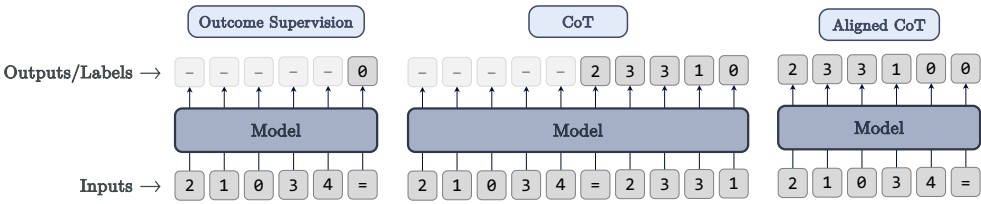

Figure 1: The three task formats for the addition modulo 5 task applied to the sequence 2 1 0 3 4.

## 3 WEIGHT SHARING ACROSS SEQUENCE LENGTH

A key hypothesis for why recurrent networks dominate transformers with respect to data efficiency, as shown in the previous section, is that their "induction bias" encourages step-by-step updates to their representations of state. This, in turn, should allow the model to share the same solution mechanisms across the whole sequence length. In this section, we investigate the extent to which the learned mechanisms are shared across different sequence lengths. Specifically, we examine whether the model develops length-specific heuristics, effectively "specialized circuits" for fixed-length sequences, or whether it has internalized the inherent inductive structure of the task.

We quantify the cross-length mechanism sharing through the lens of sample efficiency (See Section C.1 for a comprehensive discussion on data-efficiency results). Intuitively, if a model utilizes a shared mechanism (e.g., a transition operator) across varying lengths, the sample cost to learn the task over a distribution of lengths should be significantly lower than the sum of costs to learn each length individually. This is due to the *amortization* of the learning cost: the data required to learn the operation at length $n$ simultaneously contributes to the model's learning at length $n + k$.

Formally, we compare the total number of training examples required for a model to simultaneously learn the task for all sequence lengths $n \in \{2, \ldots, L\}$ (the joint task) against the sum of samples required by $L - 1$ independent models, each optimized for a single fixed length. Let $N^*_{\text{joint}}$ denote the minimal sample size required for the joint task, and $N^*_n$ denote the minimal sample size for a model trained and evaluated exclusively on sequences of length $n$. We define the *Sharing Factor* $\kappa$ as:

$$\kappa = \frac{\sum_{n=2}^{L} N^*_n}{N^*_{\text{joint}}} \tag{2}$$

The value of $\kappa$ provides insight into the extent of across-length mechanism sharing:

- $\kappa > 1$ indicates mechanism sharing and amortized learning. This suggests the model has internalized the inductive nature of the task, and data from one sequence length accelerates the acquisition of the task across the entire distribution.
- $\kappa \approx 1$ suggests that the model learns length-specific solutions in isolation, effectively partitioning capacity into independent circuits.
- $\kappa < 1$ represents a regime of destructive interference. In this case, the length-specific solutions compete for model capacity, making it more data-efficient to train separate models for each length than to optimize a single model for the joint task.

Figure 2 illustrates the sample complexity to learn the task for all sequence lengths ($N^*_{\text{joint}}$) compared against the cumulative samples required by independent models trained on individual sequence lengths ($\sum_{n=2}^{L} N^*_n$), for $L \in \{2, \ldots, 10\}$. We evaluate these metrics for modular addition with $m = 5$ across the three previously defined task formats, and draw the following key observations. Comparable results for the permutation composition task (symmetric group $S_5$) are reported in Appendix Section C.3.

**Observation 3.1.** *Transformers have low sharing factor for all task formats.*

As demonstrated, we observe a low sharing factor in transformers across all task formats, with $\kappa \approx 1$ or $\kappa < 1$ in all cases. Notably, in the Chain-of-Thought (CoT) setting, despite being transformer's most efficient task configuration, we observe an extreme case of length isolation ($\kappa = 0.28$).

**Observation 3.2.** *Transformers show destructive interference with CoT.*

The observed sharing factor of $\kappa \ll 1$ for transformer with CoT indicates a regime of destructive interference where length-specific solutions compete for model capacity, such that training on a diverse length distribution is substantially less data-efficient than training independent models on each length.

**Observation 3.3.** *Recurrent networks have high sharing factors in their preferred task formats.*

Both recurrent models exhibit clear evidence of mechanism sharing and amortized learning across sequence lengths ($\kappa \gg 1$) under the Outcome Supervision and Aligned Chain-of-Thought formats. However, this is no longer the case in the Chain-of-Thought format ($\kappa \approx 1$), where the recurrent models also fail to share across the sequence lengths, likely due to the previously discussed recall bottleneck. However, unlike transformers, we do not observe destructive interference in this case.

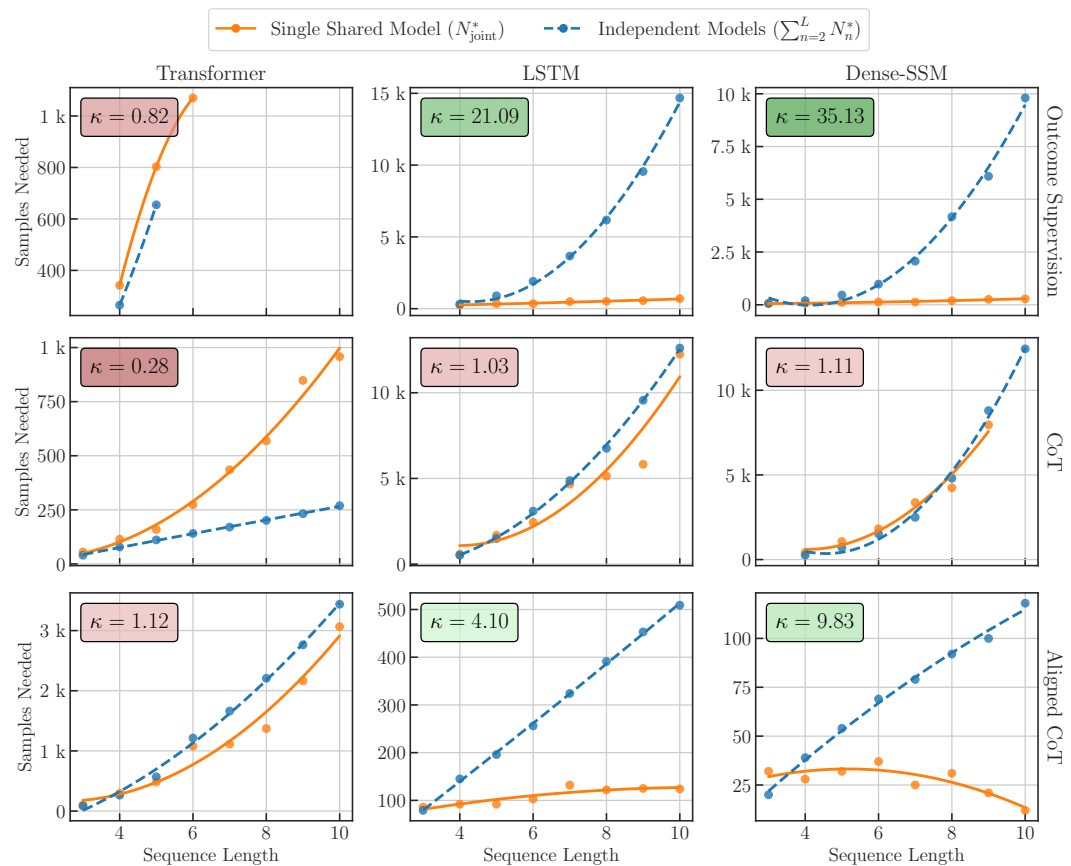

Figure 2: Sample complexity comparison between training a single model jointly across all sequence lengths and the cumulative sample complexity of independently trained models for each sequence length, together with the corresponding sharing factor. The results suggest that transformers learn largely isolated solutions for each sequence length.

**Observation 3.4.** *Longer sequences increase data efficiency for Dense-SSMs.*

As noted in the previous section, the sample requirement for the Dense-SSM under ACoT *decreases* as the maximum sequence length $L$ increases. This indicates that through cross-length mechanism sharing, the model leverages the higher density of supervision signals in longer sequences.

**Observation 3.5.** *OOD generalization implies high sharing factor, and vice versa.*

Interestingly, we observe a consistent correlation between the sharing factor $\kappa$ and length generalization: cases with high sharing factor ($\kappa \gg 1$) correspond to those in which the model learns a length-generalizable solution (see Table 2). Conversely, cases with low sharing factor ($\kappa \leq 1$) are precisely those in which the learned solution fails to extrapolate beyond the training sequence lengths. This provides additional evidence that in-distribution data efficiency and circuit sharing are fundamental implications of length generalization in state tracking.

## 4    CONCLUSIONS

Our study indicates that state tracking poses severe challenges for transformer-based sequence models not only out-of-distribution but also in-distribution: They require extraordinarily large amounts of training data to generalize on simple tasks and require Chain-of-Thought supervision to learn in-distribution on even moderate sequence lengths. This suggests that end-to-end learning in applied "agentic" scenarios, such as robotics or GUI control, could be even more challenging. The fact that data requirements scale with sequence length may also help explain well-known challenges at large context lengths ("context rot").

## SCIENCE OF DL IMPROVEMENT CHALLENGE SUBMISSION

### WHAT MODEL ARE YOU TARGETING?

*Provide a summary of the problem the deep net model is designed to solve. Good summaries should outline the state of the literature, provide an overview that domain experts would consider reasonable, and cite relevant sources.*

We target transformer-based sequence models and their fundamental limitations in performing sequential state tracking, which is the ability to maintain and manipulate a persistent hidden state over time. While transformers have achieved dominance in natural language processing, a growing body of literature has highlighted their inability to generalize to sequences longer than those seen during training (length extrapolation). Current research primarily characterizes this as an out-of-distribution (OOD) failure. We investigate whether transformers are simply failing to extrapolate, or whether they are failing to learn reusable state-tracking mechanisms even within the training distribution. Therefore, our work shows significant implications of this limitation for the in-distribution case.

### HOW DO YOUR RESULTS CONTRIBUTE—OR COULD POTENTIALLY CONTRIBUTE—TO UNDERSTANDING THESE MODELS?

*What aspects of the models become better understood thanks to your work?*

Our results fundamentally reframe the "length generalization" problem as a sample-efficiency bottleneck. We contribute three key insights into the Transformer architecture:

- *Mechanism isolation vs. reuse:* We show that the inability to extrapolate is a symptom of a deeper issue. By introducing a novel metric, the *Sharing Factor*, we demonstrate that transformers suffer from mechanism isolation. Unlike RNNs, which exhibit amortized learning (where data from one sequence length improves performance on others), transformers tend to learn disjoint, length-specific solutions. In some cases, we even observe destructive interference, where length-specific solutions compete for model capacity, so that training on a diverse length distribution is substantially less data-efficient than training separate models for each length.

- *Data inefficiency:* As a result, transformers require orders of magnitude more data than RNNs to learn simple state-tracking tasks, with sample complexity scaling sharply with sequence length and state-space size.

- *Induction bias and long-horizon implications:* We find that effective state tracking benefits from an induction bias toward step-by-step state updates and parameter sharing across sequence lengths. The absence of this bias in transformers helps explain their weak transfer across lengths and their reliance on stronger supervision (for example, step-by-step or Chain-of-Thought signals) as well as substantially larger training datasets. These trends have direct implications for long-horizon, agentic interaction settings, and they offer a fresh perspective on failures at long context lengths.

### HOW DO YOU EXPECT YOUR SUBMISSION TO INFLUENCE FUTURE WORK?

*Propose ways in which your insights, findings, or methodologies could shape subsequent research directions, model design choices, or scientific applications.*

Our findings suggest an inherent generalization limitation of self-attention that is currently being remedied only partially by scaling data or model size. We anticipate that this work could help shape future research in architectural design by motivating the incorporation of *induction bias*, including recurrent primitives, and by elevating *mechanism reuse* (for example, shared state-update rules across time) as a prerequisite for robust generalization.

More broadly, our results show that these limitations matter not only for OOD length extrapolation but also *in-distribution*. Data requirements grow sharply with sequence length and state-space size, and transformers exhibit weak (or even negative) transfer across lengths. This has direct implications for *agentic* settings, such as robotics, tool use, and GUI control, where reliable step-by-step state updates over long interaction horizons are essential and where collecting exhaustive, length-covering

supervision is costly. Finally, the observed length-dependent data inefficiency and failure to amortize learning across timesteps provides a concrete lens on challenges at long context lengths ("context rot"), motivating research on architectures and training methods that encourage incremental state updates, durable state memory, and length-agnostic generalization.

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

# A  IMPLEMENTATION DETAILS

## A.1  SEARCH PROCEDURE FOR DETERMINING $N^*$

To identify the minimal sample size $N^*$ required for a model to successfully learn a target task, we use a hybrid Binary-Geometric search as described in Algorithm 1. The algorithm conducts a search over sample sizes, combining an initial exponential reduction phase with a subsequent binary search phase.

The search begins at a predefined maximum sample size $N_{\mathrm{max}}$. For any candidate size $N$, the algorithm trains models using multiple configurations drawn from a fixed hyperparameter grid $\Phi$. In our implementation, each evaluation consists of 15 model instances (3 learning rates $\times$ 5 random seeds). A sample size $N$ is considered successful if at least one configuration attains validation loss below a threshold $\epsilon$, in which case we decrease the next sample size, and otherwise, the size is labeled unsuccessful and the next trial size is increased.

---

**Algorithm 1** Binary–Geometric Search for $N^*$

---

**Inputs:** Max sample size $N_{\mathrm{max}}$, Geometric Multiplier $M$, Step limit $S$, Hyperparameter grid $\Phi$, threshold $\epsilon$

**Output:** Minimal sample size $N^*$

1: $L \leftarrow 0$          *# Lower bound (largest failed size tried so far)*
2: $N^* \leftarrow N_{\mathrm{max}}$        *# Best size so far (smallest successful size tried so far)*
3: $N \leftarrow N_{\mathrm{max}}$         *# Current candidate*
4: $step \leftarrow 0$

5: **while** $step < S$ **do**

6:   *# Step 1: Evaluate model configuration at size $N$*
7:   $Success \leftarrow$ **false**
8:   **for all** $\phi \in \Phi$ **do**
9:    $\mathcal{L}_{\mathrm{val}} \leftarrow$ Train and evaluate model with $\phi$ on $N$ samples
10:    **if** $\mathcal{L}_{\mathrm{val}} < \epsilon$ **then**
11:     $Success \leftarrow$ **true**
12:     **break**

13:   *# Step 2: Update bounds & choose next candidate*
14:   **if** $Success$ **then**
15:    $N^* \leftarrow N$       *# Update the best value*
16:    **if** $L = 0$: $N \leftarrow N \mathbin{//} M$   *# Phase 1: Exponential decay*
17:    **else:** $N \leftarrow (L + N) \mathbin{//} 2$   *# Phase 2: Binary search update*
18:   **else**
19:    **if** $N = N_{\mathrm{max}}$ **then**
20:     **return** $-1$       *# Failure: task not learned with max sample size*
21:    $L \leftarrow N$        *# Update lower bound*
22:    $N \leftarrow (N + N^*) \mathbin{//} 2$    *# Binary search update*

23:   $step \leftarrow step + 1$

24: **return** $N^*$

---

We use a geometric multiplier of $M = 1000$, a maximum of $S = 20$ search steps, and a success threshold of $\epsilon = 10^{-4}$. The hyperparameter grid is

$$\Phi = \{\mathrm{LR} \in \{10^{-3}, 10^{-4}, 10^{-5}\}\} \times \{\mathrm{seed} \in \{10, 20, 30, 40, 50\}\},$$

yielding 15 configurations per evaluation. Each model is trained for a fixed budget of 250k optimization steps with batch size 64 using the Adam optimizer Kingma & Ba (2014), independent of the training set size $N$. This implies a maximum feasible sample size of

$$N_{\mathrm{max}} = 250{,}000 \times 64 = 16\mathrm{M}.$$

The maximum training set size is the minimum between the feasible 16M samples and the total number of sequences available under the specified configuration of maximum sequence length $L$

and modulus $m$: $m^L$ for the *Fixed* distribution, and $\sum_{n=1}^{L} m^n$ for the uniform and short-to-long distributions (see Section 2). The final training set size is obtained after deducting the validation set. We perform over *190,000 training runs* for the results reported in this paper, excluding development runs.

## A.2 EVALUATION

We ensure that the training and validation sets are strictly disjoint. The validation set contains 2,000 samples (or at most 20% of the available data) and remains identical across different training set sizes, except for variations introduced by the random seed. In addition, we always use at most 20% of the available samples at each sequence length for validation, with the remainder reserved exclusively for training. Also, for all tasks, multi-digit integers are represented as single tokens during tokenization. Finally, for the Chain-of-Thought task format, validation loss is computed using teacher forcing rather than autoregressive sampling.

## A.3 MODELS

The transformer model is based on the GPT-2 architecture (Radford et al., 2019), with 6 layers and a model (embedding/hidden) dimension of 256. Other architectural parameters, including an MLP expansion factor of 4, follow the default GPT-2 (small) settings.

Both the LSTM and Dense-SSM use a single-layer recurrent cell followed by a linear classification head to map the hidden state to token logits. We use an input and hidden dimension of 768 for the LSTM, and 256 for the Dense-SSM. See Ebrahimi & Memisevic (2025) for additional details on the Bilinear architecture used for Dense-SSM. We also experiment with a 2-layer transformer and a single-layer LSTM with a hidden dimensionality of 256; sample-efficiency results for these variants are provided in the Appendix Section C.4.

## B RELATED WORK

A range of studies has shown that transformer-based sequence models fail to length-generalize in state tracking tasks Anil et al. (2022b); Deletang et al. (2023); Dziri et al. (2024); Abbe et al. (2024); Ebrahimi et al. (2024). Unlike our work, these studies solely discuss OOD scenarios, while we discuss in-distribution data efficiency instead.

The inability to length-generalize in state tracking tasks has been shown to hold also for most existing state-space models (SSM) Sarrof et al. (2024); Merrill et al. (2024). However recent work has shown that making the hidden-to-hidden transition matrix in the SSM input-dependent and non-diagonal can recover the ability to length-generalize. Fan et al. (2024); Grazzi et al. (2025); Ebrahimi & Memisevic (2025); Beck et al. (2024).

Liu et al. (2023); Li et al. (2025) show that transformers solve state tracking tasks in-distribution by making use of parallel mechanisms reminiscent of associative scan. While this view can help explain the OOD failures of these models, it also hints at the absence of an "induction" bias which affects data efficiency as we show in this work.

## C ADDITIONAL EXPERIMENTAL RESULTS

### C.1 IN-DISTRIBUTION DATA EFFICIENCY

We perform the above binary search procedure to identify the minimal dataset size ($N^*$) across all combinations of maximum sequence length $L \in \{5, 10, 20, 30\}$ and modulus $m \in \{2, 3, 5, 10, 15, 20, 50, 75, 100\}$, for each of the three task formats, length distributions, and models described earlier. The results are summarized in Table 1. For ease of comparison, we also visualize selected slices of this table in figures throughout this section. Comparable results for models of different sizes are reported in Appendix Section C.4. From the table we can infer the following key observations:

**Observation C.1.** *Transformers prefer non-aligned supervision (Chain of Thought).*

We observe a clear preference of transformers for CoT over the Aligned CoT format. For example, at $m = 5$ and $L = 20$, CoT requires 1.7K samples, while Aligned CoT requires 2M, an order-of-magnitude increase in sample complexity. Figure 3 further illustrates this gap in sample requirements for the case $m = 2$ (parity) across the two formats.

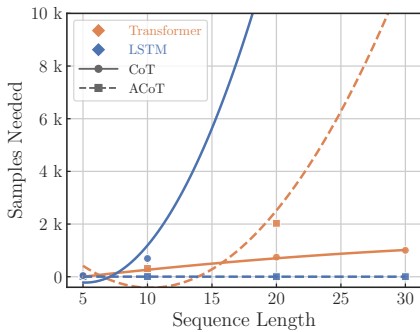

It has been hypothesized that by outputting intermediate steps autoregressively, the model can attend to its own previous outputs, effectively simulating a larger depth circuit Li et al. (2024), and the results confirm this hypothesis. In contrast, Aligned Chain-of-Thought forces the model to compress the computation into a single forward pass per token without the benefit of re-attending to intermediate results, which appears less aligned with the transformer's non-recurrent nature.

Figure 3: Minimal dataset size for the uniform length distribution with $m = 2$.

**Observation C.2.** *Recurrent models prefer aligned supervision (Aligned Chain-of-Thought).*

Conversely, recurrent models (LSTMs and Dense-SSMs) demonstrate superior sample efficiency when trained with the Aligned CoT (ACoT) format, which provides supervision aligned with the evolution of the hidden state (see also Figure 3).

In contrast, RNNs struggle with CoT, which is likely due to their recall bottleneck: a model must output the sequence of partial sums $(s_1, \ldots, s_n)$ *after* processing the entire input sequence. This effectively requires it to unroll the chain of intermediate computations from the beginning. In fact, we note that under the CoT format, recurrent models even fail to generalize to longer sequences despite their sequential inductive bias (see Table 2 for length-generalization results). The task thereby becomes bottlenecked by the model's limited memory capacity rather than its state-tracking ability.

**Observation C.3.** *Recurrent models outperform transformers in the absence of intermediate supervision.*

In the Outcome Supervision setting, the model must implicitly infer the latent algebraic structure of the task solely from the final solution, without any guidance on the intermediate steps. This requires the model to effectively marginalize over unobserved computational paths with difficulty scaling with both the state space size $m$ and sequence length $n$.

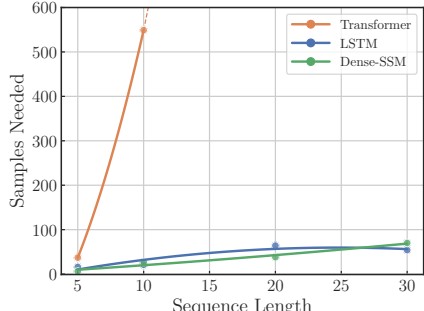

We observe that recurrent models significantly outperform transformers in this regime. While transformers fail to converge for all but the most trivial configurations (very small $m$ and $n$), the recurrent architectures successfully learn the task for higher moduli and extended sequence lengths, achieving convergence with orders of magnitude fewer training samples. Figure 4 illustrates this behavior for the parity case ($m = 2$).

Figure 4: $N^*$ for the outcome supervision format with a uniform length distribution and $m = 2$ (parity).

**Observation C.4.** *With intermediate supervision, longer sequences improve the data efficiency of recurrent models but not transformers.*

Intuitively, under formats with intermediate supervision (CoT or ACoT), longer sequences should improve sample efficiency. This is because with intermediate solutions, the effective amount of supervised tokens increases linearly with sequence length.

We validate this hypothesis in recurrent models trained with Aligned Chain-of-Thought: the fixed length distribution (comprising only longest sequences) yields the highest data efficiency, followed by uniform, and finally short-to-long.

Furthermore, in the uniform setting, we find that recurrent models trained with ACoT require fewer data points as the maximum sequence length $L$ increases, as expected. In contrast, transformers trained with CoT fail to leverage this additional supervision. This trend is also evident in Figure 5, which compares sample complexity on the parity task.

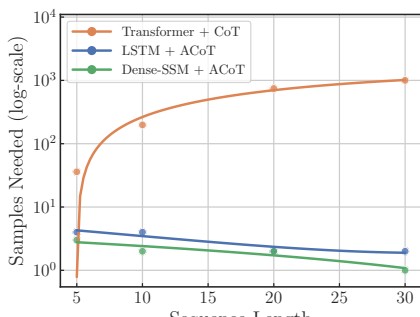

Figure 5: Sample complexity (log scale) for transformers trained with CoT and RNNs trained with ACoT on the parity task.

**Observation C.5.** *With outcome supervision, short sequences are more valuable for recurrent models.*

In the Outcome Supervision setting, we compare the data requirements under the uniform and short-to-long length distributions. Recall that these two distributions differ only in the order in which samples are presented during training.

We observe that recurrent models require fewer samples in the short-to-long setting, suggesting that shorter sequences provide a stronger learning signal than longer sequences for these models. This effect is illustrated in Figure 6 for the case $m = 2$.

Altogether, these observations provide evidence that the key difference enabling data efficiency for recurrent networks, and even more so the Dense-SSM models, is their tendency to represent state transitions inherent in the data generation process at every time-step.

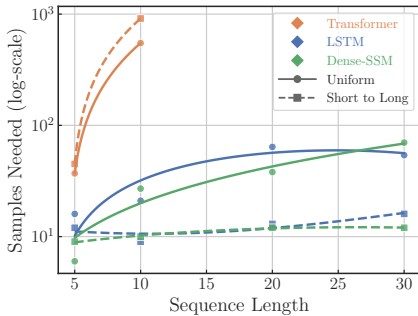

Figure 6: Sample complexity (log scale) in the Outcome Supervision format for the uniform and short-to-long settings, with $m = 2$.

### C.2 EVALUATING LENGTH-GENERALIZATION

Table 2 reports accuracy on sequences of length $2\times$ the maximum length used during training, normalized such that $0$ corresponds to random chance. All models are trained using the maximum available training set size for each configuration.

Table 1: Minimal number of training samples required to learn the modulo addition task. A dash (–) indicates that the task was not learned at the maximum training set size.

| Model | Format | Mod. | Fixed | | | | Uniform | | | | Short-to-Long | | | |
|---|---|---|---|---|---|---|---|---|---|---|---|---|---|---|
| | | | 5 | 10 | 20 | 30 | 5 | 10 | 20 | 30 | 5 | 10 | 20 | 30 |
| Transformer | Outcome Supervision | 2 | 19 | 364 | – | – | 37 | 549 | – | – | 45 | 913 | – | – |
| | | 3 | 119 | – | – | – | – | – | – | – | 243 | – | – | – |
| | | 5 | 1.1K | – | – | – | 2.6K | – | – | – | 1.5K | – | – | – |
| | | 10 | – | – | – | – | – | – | – | – | – | – | – | – |
| | | 15 | – | – | – | – | – | – | – | – | – | – | – | – |
| | | 20 | – | – | – | – | – | – | – | – | – | – | – | – |
| | | 50 | – | – | – | – | – | – | – | – | – | – | – | – |
| | | 75 | – | – | – | – | – | – | – | – | – | – | – | – |
| | | 100 | – | – | – | – | – | – | – | – | – | – | – | – |
| | CoT | 2 | 10 | 16 | 19 | 20 | 36 | 198 | 744 | 1K | 39 | 824 | 1M | – |
| | | 3 | 16 | 18 | 21 | 25 | 78 | 465 | 1.2K | 1.4K | 108 | 28.8K | – | – |
| | | 5 | 33 | 30 | 31 | 34 | 148 | 1.1K | 1.7K | 1.7K | 647 | 2.4M | – | – |
| | | 10 | 94 | 66 | 62 | 70 | 427 | 2.3K | 3.4K | 4K | 11K | – | – | – |
| | | 15 | 166 | 116 | 78 | 107 | 709 | 2.5K | 5K | 5.1K | 54.3K | – | – | – |
| | | 20 | 377 | 178 | 116 | 135 | 1K | 3.8K | 5.6K | 7.4K | 168.6K | – | – | – |
| | | 50 | 1.1K | 678 | 553 | 470 | 2.6K | 8.6K | 12K | 11.2K | 6.4M | – | – | – |
| | | 75 | 1.7K | 1.2K | 1K | 946 | 4.2K | 11K | 18.1K | 16.1K | – | – | – | – |
| | | 100 | 2.5K | 1.9K | 1.6K | 1.6K | 5.8K | 13.2K | 21.1K | 19.1K | – | – | – | – |
| | Aligned CoT | 2 | 16 | 121 | 644 | 1.2K | 27 | 313 | 2K | 11.7K | 19 | 856 | 1M | – |
| | | 3 | 91 | 2K | 3.7K | 4K | 180 | 8.3K | 66.8K | – | 207 | 29.7K | – | – |
| | | 5 | 528 | 3.9K | 9.2K | 20.8K | 671 | 14.5K | 2M | – | 909 | 2.4M | – | – |
| | | 10 | 1.3K | 6K | 15.5K | 131.4K | 1.7K | 32.3K | – | – | 11.2K | – | – | – |
| | | 15 | 1.5K | 7.9K | 23.3K | – | 2.4K | 61.5K | – | – | 55K | – | – | – |
| | | 20 | 2.6K | 21.9K | 85.3K | – | 4.5K | 190.9K | – | – | 172.6K | – | – | – |
| | | 50 | 13K | 1M | – | – | 22K | – | – | – | 6.7M | – | – | – |
| | | 75 | 21.1K | 15.2M | – | – | 102.1K | – | – | – | – | – | – | – |
| | | 100 | 23.6K | – | – | – | 8M | – | – | – | – | – | – | – |
| LSTM | Outcome Supervision | 2 | 14 | 101 | 252 | 309 | 16 | 21 | 64 | 54 | 12 | 9 | 13 | 16 |
| | | 3 | 195 | 620 | 1.7K | – | 78 | 122 | 317 | 526 | 83 | 90 | 90 | 100 |
| | | 5 | 2K | 5.9K | – | – | 387 | 707 | – | – | 381 | 22.7K | – | – |
| | | 10 | 6.4K | – | – | – | 2.2K | 8.6K | – | – | 2.2K | – | – | – |
| | | 15 | 18.8K | – | – | – | 11.8K | – | – | – | – | – | – | – |
| | | 20 | – | – | – | – | – | – | – | – | – | – | – | – |
| | | 50 | – | – | – | – | – | – | – | – | – | – | – | – |
| | | 75 | – | – | – | – | – | – | – | – | – | – | – | – |
| | | 100 | – | – | – | – | – | – | – | – | – | – | – | – |
| | CoT | 2 | 20 | 307 | 4.7K | 8.6K | 45 | 692 | 13.4K | 37.1K | 48 | 843 | 1.1M | – |
| | | 3 | – | 2.9K | 9.3K | 14K | – | 7.4K | 40.3K | 5.2M | – | 29K | – | – |
| | | 5 | 1.6K | 10.3K | 20.4K | 44K | 2K | 28.4K | 68.1K | 2.1M | 2.1K | – | – | – |
| | | 10 | 6.1K | 15.7K | 29.1K | – | 14.3K | 544.4K | 500K | – | 16.8K | – | – | – |
| | | 15 | 15.8K | 21.1K | 42K | – | 32K | 67K | – | – | 65.6K | – | – | – |
| | | 20 | 21.2K | 29.6K | 54.3K | – | 40.7K | 409.1K | 14M | – | 215.4K | – | – | – |
| | | 50 | 43K | 50.4K | – | – | 206K | 2.2M | – | – | 10.4M | – | – | – |
| | | 75 | 56.5K | 58K | – | – | 4M | – | – | – | – | – | – | – |
| | | 100 | 84.2K | 9.5M | – | – | – | – | – | – | – | – | – | – |
| | Aligned CoT | 2 | 4 | 6 | 2 | 2 | 4 | 4 | 2 | 2 | 9 | 7 | 7 | 8 |
| | | 3 | 10 | 8 | 8 | 6 | 18 | 12 | 6 | 8 | 27 | 28 | 34 | 34 |
| | | 5 | 95 | 70 | 39 | 31 | 153 | 107 | 80 | 47 | 142 | 148 | 253 | 235 |
| | | 10 | 278 | 194 | 146 | 130 | 447 | 313 | 247 | 232 | 685 | 1.3K | 1.8K | 2K |
| | | 15 | 534 | 478 | 409 | 371 | 865 | 631 | 567 | 500 | 1.5K | 3.9K | 5.2K | 7.1K |
| | | 20 | 798 | 672 | 652 | 798 | 1.3K | 934 | 870 | 820 | 2.7K | 9.4K | 13.4K | 25.9K |
| | | 50 | 3.7K | 4.6K | 6K | 7.4K | 4.1K | 5.3K | 6.1K | 6.9K | 14.7K | 156.2K | – | – |
| | | 75 | 7.4K | 9.2K | 12.2K | 13.3K | 8.4K | 11.3K | 14.2K | 16.8K | 41.1K | 697.4K | – | – |
| | | 100 | 12.8K | 16.6K | 21.9K | 23.7K | 13.3K | 19.4K | 24.3K | 28.5K | 99.7K | – | – | – |
| Dense-SSM | Outcome Supervision | 2 | 13 | 58 | 33 | 70 | 6 | 27 | 38 | 70 | 9 | 10 | 12 | 12 |
| | | 3 | 56 | 390 | 8K | 2.2K | 31 | 69 | 209 | 265 | 24 | 32 | 41 | 41 |
| | | 5 | 323 | 3.8K | 10.9M | – | 118 | 285 | 1K | 1.1K | 69 | 83 | 132 | 163 |
| | | 10 | 2.7K | 8.3M | – | – | 573 | 2.2K | 4.5K | – | 321 | 452 | 678 | 1.1K |
| | | 15 | 6.5K | – | – | – | 1.6K | – | 12.1K | – | 970 | 1.4K | 3.7K | 3.8K |
| | | 20 | 15.7K | – | – | – | 3K | – | – | – | 2K | 8.4K | 8.8K | 9K |
| | | 50 | – | – | – | – | 15.6K | – | – | – | 11K | – | – | – |
| | | 75 | – | – | – | – | – | – | – | – | – | – | – | – |
| | | 100 | – | – | – | – | – | – | – | – | – | – | – | – |
| | CoT | 2 | – | 172 | 2.1K | – | – | 511 | – | – | – | 875 | – | – |
| | | 3 | 110 | 1K | – | – | 216 | 6.2K | – | – | 234 | 29.8K | – | – |
| | | 5 | 538 | 5.9K | – | – | 1.5K | 19.5K | – | – | 1.5K | 2.4M | – | – |
| | | 10 | 3.4K | – | – | – | 12.5K | – | – | – | 14.8K | – | – | – |
| | | 15 | 12.6K | – | – | – | 21.2K | – | – | – | 56.2K | – | – | – |
| | | 20 | 21.9K | – | – | – | – | – | – | – | 2.5M | – | – | – |
| | | 50 | – | – | – | – | – | – | – | – | – | – | – | – |
| | | 75 | – | – | – | – | – | – | – | – | – | – | – | – |
| | | 100 | – | – | – | – | – | – | – | – | – | – | – | – |
| | Aligned CoT | 2 | 2 | 2 | 3 | 1 | 3 | 2 | 2 | 1 | 6 | 4 | 4 | 4 |
| | | 3 | 4 | 6 | 4 | 4 | 7 | 6 | 4 | 4 | 9 | 11 | 10 | 10 |
| | | 5 | 18 | 9 | 6 | 6 | 24 | 17 | 10 | 10 | 30 | 26 | 25 | 25 |
| | | 10 | 109 | 41 | 21 | 17 | 148 | 74 | 56 | 27 | 102 | 101 | 101 | 101 |
| | | 15 | 219 | 109 | 66 | 37 | 412 | 186 | 101 | 70 | 228 | 225 | 225 | 225 |
| | | 20 | 438 | 233 | 124 | 78 | 633 | 382 | 241 | 116 | 396 | 397 | 405 | 405 |
| | | 50 | 3.2K | 1.7K | 1.1K | 874 | 4.5K | 3.2K | 2K | 1.6K | 2.5K | 2.5K | 2.5K | 2.5K |
| | | 75 | 8.1K | 4K | 3.1K | 2.1K | 10.8K | 6.8K | 5.1K | 3.8K | 5.6K | 5.6K | 5.6K | 5.6K |
| | | 100 | 12.5K | 6K | 4.3K | 3K | 16.1K | 10.2K | 6.7K | 6K | 10K | 10K | 12.8K | 16.4K |

Table 2: Accuracy on sequences of length $2\times$ the maximum used during training, normalized such that 0 corresponds to random chance.

| Model | Format | Modulus | Fixed 5 | 10 | 20 | 30 | Uniform 5 | 10 | 20 | 30 | Short-to-Long 5 | 10 | 20 | 30 |
|---|---|---|---|---|---|---|---|---|---|---|---|---|---|---|
| Transformer | Outcome Supervision | 2 | 0.00 | 0.00 | 0.01 | 0.00 | 0.01 | 0.00 | 0.00 | 0.00 | 0.00 | 0.00 | 0.00 | 0.00 |
| | | 3 | 0.01 | 0.00 | 0.01 | 0.00 | 0.00 | 0.04 | 0.00 | 0.00 | 0.00 | 0.03 | 0.00 | 0.00 |
| | | 5 | 0.00 | 0.00 | 0.00 | 0.00 | 0.00 | 0.03 | 0.01 | 0.00 | 0.00 | 0.00 | 0.00 | 0.00 |
| | | 10 | 0.00 | 0.02 | 0.00 | 0.00 | 0.00 | 0.00 | 0.01 | 0.00 | 0.01 | 0.00 | 0.00 | 0.00 |
| | | 15 | 0.00 | 0.01 | 0.00 | 0.00 | 0.00 | 0.00 | 0.01 | 0.00 | 0.01 | 0.01 | 0.01 | 0.00 |
| | | 20 | 0.00 | 0.00 | 0.01 | 0.00 | 0.00 | 0.01 | 0.01 | 0.00 | 0.00 | 0.00 | 0.00 | 0.00 |
| | | 50 | 0.01 | 0.00 | 0.00 | 0.00 | 0.00 | 0.00 | 0.01 | 0.00 | 0.00 | 0.00 | 0.00 | 0.00 |
| | | 75 | 0.00 | 0.00 | 0.00 | 0.00 | 0.00 | 0.00 | 0.00 | 0.00 | 0.00 | 0.00 | 0.00 | 0.00 |
| | | 100 | 0.00 | 0.00 | 0.00 | 0.00 | 0.00 | 0.00 | 0.00 | 0.00 | 0.00 | 0.00 | 0.00 | 0.00 |
| | Chain of Thought | 2 | 0.00 | 0.01 | 0.00 | 0.02 | 0.00 | 0.00 | 0.00 | 0.00 | 0.00 | 0.00 | 0.00 | 0.00 |
| | | 3 | 0.02 | 0.02 | 0.01 | 0.00 | 0.00 | 0.00 | 0.00 | 0.02 | 0.04 | 0.02 | 0.02 | 0.00 |
| | | 5 | 0.01 | 0.00 | 0.00 | 0.01 | 0.00 | 0.00 | 0.01 | 0.01 | 0.00 | 0.00 | 0.00 | 0.00 |
| | | 10 | 0.01 | 0.01 | 0.01 | 0.01 | 0.01 | 0.00 | 0.00 | 0.00 | 0.00 | 0.00 | 0.00 | 0.00 |
| | | 15 | 0.01 | 0.00 | 0.01 | 0.00 | 0.00 | 0.01 | 0.01 | 0.01 | 0.00 | 0.00 | 0.00 | 0.01 |
| | | 20 | 0.00 | 0.00 | 0.00 | 0.00 | 0.00 | 0.00 | 0.00 | 0.01 | 0.00 | 0.00 | 0.01 | 0.00 |
| | | 50 | 0.00 | 0.00 | 0.00 | 0.00 | 0.00 | 0.00 | 0.00 | 0.00 | 0.00 | 0.00 | 0.00 | 0.00 |
| | | 75 | 0.00 | 0.00 | 0.00 | 0.00 | 0.01 | 0.00 | 0.00 | 0.00 | 0.01 | 0.00 | 0.00 | 0.00 |
| | | 100 | 0.00 | 0.00 | 0.00 | 0.00 | 0.00 | 0.00 | 0.00 | 0.00 | 0.00 | 0.00 | 0.00 | 0.00 |
| | Aligned Chain-of-Thought | 2 | 0.00 | 0.00 | 0.00 | 0.00 | 0.01 | 0.00 | 0.00 | 0.00 | 0.01 | 0.02 | 0.01 | 0.00 |
| | | 3 | 0.00 | 0.01 | 0.00 | 0.00 | 0.00 | 0.00 | 0.03 | 0.00 | 0.00 | 0.01 | 0.02 | 0.00 |
| | | 5 | 0.00 | 0.01 | 0.01 | 0.00 | 0.00 | 0.01 | 0.00 | 0.01 | 0.00 | 0.01 | 0.00 | 0.00 |
| | | 10 | 0.00 | 0.01 | 0.00 | 0.00 | 0.00 | 0.00 | 0.00 | 0.00 | 0.01 | 0.01 | 0.01 | 0.00 |
| | | 15 | 0.01 | 0.00 | 0.01 | 0.00 | 0.01 | 0.00 | 0.01 | 0.01 | 0.00 | 0.00 | 0.01 | 0.00 |
| | | 20 | 0.01 | 0.00 | 0.00 | 0.00 | 0.01 | 0.01 | 0.00 | 0.00 | 0.00 | 0.00 | 0.00 | 0.00 |
| | | 50 | 0.00 | 0.00 | 0.00 | 0.00 | 0.00 | 0.00 | 0.00 | 0.00 | 0.00 | 0.00 | 0.00 | 0.00 |
| | | 75 | 0.00 | 0.00 | 0.00 | 0.00 | 0.00 | 0.00 | 0.00 | 0.00 | 0.00 | 0.00 | 0.00 | 0.00 |
| | | 100 | 0.00 | 0.01 | 0.00 | 0.00 | 0.00 | 0.00 | 0.00 | 0.00 | 0.00 | 0.00 | 0.00 | 0.00 |
| LSTM | Outcome Supervision | 2 | 0.09 | 1.00 | 0.60 | 0.00 | 1.00 | 1.00 | 1.00 | 0.02 | 1.00 | 1.00 | 0.99 | 0.00 |
| | | 3 | 0.05 | 0.00 | 0.12 | 0.02 | 1.00 | 1.00 | 0.96 | 0.99 | 1.00 | 0.98 | 0.18 | 0.29 |
| | | 5 | 0.61 | 0.01 | 0.01 | 0.00 | 0.88 | 0.93 | 0.84 | 1.00 | 0.98 | 0.00 | 0.00 | 0.01 |
| | | 10 | 0.13 | 0.01 | 0.01 | 0.01 | 0.86 | 0.59 | 0.99 | 0.13 | 0.26 | 0.12 | 0.12 | 0.12 |
| | | 15 | 0.92 | 0.00 | 0.01 | 0.00 | 0.50 | 0.11 | 0.01 | 0.01 | 0.17 | 0.06 | 0.16 | 0.02 |
| | | 20 | 0.00 | 0.00 | 0.01 | 0.00 | 0.00 | 0.00 | 0.00 | 0.00 | 0.02 | 0.00 | 0.00 | 0.00 |
| | | 50 | 0.00 | 0.00 | 0.00 | 0.00 | 0.00 | 0.00 | 0.00 | 0.00 | 0.00 | 0.00 | 0.00 | 0.00 |
| | | 75 | 0.00 | 0.00 | 0.00 | 0.00 | 0.00 | 0.00 | 0.00 | 0.00 | 0.00 | 0.00 | 0.00 | 0.01 |
| | | 100 | 0.00 | 0.00 | 0.00 | 0.00 | 0.00 | 0.00 | 0.00 | 0.00 | 0.00 | 0.00 | 0.00 | 0.00 |
| | Chain of Thought | 2 | 0.14 | 0.18 | 0.00 | 0.00 | 0.11 | 0.00 | 0.00 | 0.00 | 0.00 | 0.00 | 0.01 | 0.03 |
| | | 3 | 0.03 | 0.00 | 0.00 | 0.01 | 0.03 | 0.00 | 0.00 | 0.00 | 0.00 | 0.01 | 0.03 | 0.00 |
| | | 5 | 0.01 | 0.00 | 0.00 | 0.00 | 0.04 | 0.01 | 0.00 | 0.01 | 0.00 | 0.00 | 0.00 | 0.01 |
| | | 10 | 0.01 | 0.00 | 0.01 | 0.01 | 0.00 | 0.00 | 0.01 | 0.01 | 0.00 | 0.00 | 0.00 | 0.01 |
| | | 15 | 0.00 | 0.00 | 0.01 | 0.00 | 0.00 | 0.00 | 0.01 | 0.01 | 0.00 | 0.00 | 0.01 | 0.00 |
| | | 20 | 0.00 | 0.00 | 0.00 | 0.00 | 0.00 | 0.00 | 0.00 | 0.00 | 0.01 | 0.00 | 0.00 | 0.00 |
| | | 50 | 0.00 | 0.00 | 0.00 | 0.00 | 0.00 | 0.00 | 0.00 | 0.00 | 0.00 | 0.00 | 0.00 | 0.00 |
| | | 75 | 0.00 | 0.00 | 0.00 | 0.00 | 0.00 | 0.00 | 0.00 | 0.00 | 0.00 | 0.00 | 0.00 | 0.00 |
| | | 100 | 0.00 | 0.00 | 0.00 | 0.00 | 0.00 | 0.00 | 0.00 | 0.00 | 0.00 | 0.00 | 0.00 | 0.00 |
| | Aligned Chain-of-Thought | 2 | 1.00 | 1.00 | 1.00 | 1.00 | 1.00 | 1.00 | 1.00 | 1.00 | 1.00 | 1.00 | 1.00 | 1.00 |
| | | 3 | 1.00 | 1.00 | 1.00 | 1.00 | 1.00 | 1.00 | 1.00 | 1.00 | 1.00 | 1.00 | 1.00 | 1.00 |
| | | 5 | 1.00 | 1.00 | 1.00 | 1.00 | 1.00 | 1.00 | 1.00 | 1.00 | 1.00 | 1.00 | 1.00 | 1.00 |
| | | 10 | 1.00 | 1.00 | 1.00 | 1.00 | 1.00 | 1.00 | 1.00 | 1.00 | 1.00 | 1.00 | 1.00 | 1.00 |
| | | 15 | 1.00 | 1.00 | 1.00 | 1.00 | 1.00 | 1.00 | 1.00 | 1.00 | 1.00 | 1.00 | 1.00 | 1.00 |
| | | 20 | 1.00 | 1.00 | 1.00 | 1.00 | 1.00 | 1.00 | 1.00 | 1.00 | 1.00 | 1.00 | 1.00 | 0.59 |
| | | 50 | 1.00 | 1.00 | 1.00 | 1.00 | 1.00 | 1.00 | 1.00 | 0.99 | 1.00 | 0.98 | 0.45 | 0.14 |
| | | 75 | 1.00 | 1.00 | 0.99 | 1.00 | 1.00 | 1.00 | 1.00 | 0.99 | 0.97 | 0.16 | 0.02 | 0.02 |
| | | 100 | 1.00 | 1.00 | 0.99 | 1.00 | 1.00 | 1.00 | 0.99 | 0.98 | 0.95 | 0.16 | 0.05 | 0.03 |
| Dense-SSM | Outcome Supervision | 2 | 0.00 | 1.00 | 1.00 | 1.00 | 1.00 | 1.00 | 1.00 | 1.00 | 1.00 | 1.00 | 1.00 | 1.00 |
| | | 3 | 0.00 | 1.00 | 0.03 | 0.00 | 1.00 | 1.00 | 1.00 | 1.00 | 1.00 | 1.00 | 1.00 | 1.00 |
| | | 5 | 0.00 | 0.00 | 0.01 | 0.00 | 1.00 | 1.00 | 1.00 | 1.00 | 1.00 | 1.00 | 1.00 | 0.29 |
| | | 10 | 1.00 | 0.10 | 0.01 | 0.02 | 1.00 | 1.00 | 1.00 | 0.10 | 1.00 | 0.13 | 0.11 | 0.12 |
| | | 15 | 0.00 | 0.00 | 0.00 | 0.01 | 1.00 | 1.00 | 1.00 | 1.00 | 1.00 | 0.32 | 0.64 | 0.00 |
| | | 20 | 0.47 | 0.00 | 0.01 | 0.01 | 1.00 | 1.00 | 1.00 | 0.48 | 1.00 | 0.00 | 0.02 | 0.00 |
| | | 50 | 0.00 | 0.00 | 0.00 | 0.00 | 1.00 | 0.50 | 0.18 | 0.00 | 1.00 | 0.00 | 0.00 | 0.00 |
| | | 75 | 0.00 | 0.00 | 0.00 | 0.00 | 1.00 | 0.83 | 0.00 | 0.00 | 0.00 | 0.00 | 0.00 | 0.00 |
| | | 100 | 0.00 | 0.00 | 0.00 | 0.00 | 0.65 | 0.25 | 0.00 | 0.00 | 0.00 | 0.00 | 0.00 | 0.00 |
| | Chain of Thought | 2 | 0.00 | 0.03 | 0.01 | 0.00 | 0.00 | 0.00 | 0.00 | 0.00 | 0.00 | 0.02 | 0.02 | 0.01 |
| | | 3 | 0.02 | 0.00 | 0.01 | 0.02 | 0.00 | 0.00 | 0.00 | 0.01 | 0.02 | 0.02 | 0.00 | 0.01 |
| | | 5 | 0.00 | 0.00 | 0.00 | 0.00 | 0.00 | 0.00 | 0.00 | 0.00 | 0.00 | 0.00 | 0.00 | 0.00 |
| | | 10 | 0.01 | 0.00 | 0.00 | 0.00 | 0.00 | 0.00 | 0.00 | 0.00 | 0.00 | 0.00 | 0.00 | 0.00 |
| | | 15 | 0.00 | 0.01 | 0.01 | 0.00 | 0.00 | 0.00 | 0.00 | 0.01 | 0.00 | 0.00 | 0.00 | 0.01 |
| | | 20 | 0.00 | 0.00 | 0.00 | 0.00 | 0.00 | 0.00 | 0.00 | 0.00 | 0.00 | 0.01 | 0.01 | 0.00 |
| | | 50 | 0.00 | 0.00 | 0.00 | 0.00 | 0.00 | 0.00 | 0.00 | 0.01 | 0.00 | 0.00 | 0.00 | 0.00 |
| | | 75 | 0.00 | 0.00 | 0.00 | 0.00 | 0.00 | 0.00 | 0.00 | 0.00 | 0.00 | 0.00 | 0.00 | 0.00 |
| | | 100 | 0.00 | 0.00 | 0.00 | 0.00 | 0.00 | 0.00 | 0.00 | 0.00 | 0.00 | 0.00 | 0.00 | 0.00 |
| | Aligned Chain-of-Thought | 2 | 1.00 | 1.00 | 1.00 | 1.00 | 1.00 | 1.00 | 1.00 | 1.00 | 1.00 | 1.00 | 1.00 | 1.00 |
| | | 3 | 1.00 | 1.00 | 1.00 | 1.00 | 1.00 | 1.00 | 1.00 | 1.00 | 1.00 | 1.00 | 1.00 | 1.00 |
| | | 5 | 1.00 | 1.00 | 1.00 | 1.00 | 1.00 | 1.00 | 1.00 | 1.00 | 1.00 | 1.00 | 1.00 | 1.00 |
| | | 10 | 1.00 | 1.00 | 1.00 | 1.00 | 1.00 | 1.00 | 1.00 | 1.00 | 1.00 | 1.00 | 1.00 | 1.00 |
| | | 15 | 1.00 | 1.00 | 1.00 | 1.00 | 1.00 | 1.00 | 1.00 | 1.00 | 1.00 | 1.00 | 1.00 | 0.93 |
| | | 20 | 1.00 | 1.00 | 1.00 | 1.00 | 1.00 | 1.00 | 1.00 | 1.00 | 1.00 | 1.00 | 1.00 | 1.00 |
| | | 50 | 1.00 | 1.00 | 1.00 | 1.00 | 1.00 | 1.00 | 1.00 | 1.00 | 1.00 | 1.00 | 1.00 | 1.00 |
| | | 75 | 1.00 | 1.00 | 1.00 | 1.00 | 1.00 | 1.00 | 1.00 | 1.00 | 1.00 | 1.00 | 1.00 | 1.00 |
| | | 100 | 1.00 | 1.00 | 1.00 | 1.00 | 1.00 | 1.00 | 1.00 | 1.00 | 1.00 | 1.00 | 1.00 | 1.00 |

## C.3   PERMUTATION COMPOSITION TASK

**Task:**   To show our findings generalize beyond commutative operations, we consider the task of permutation composition (simulating the symmetric group $S_m$). Each element of the group represents a permutation of the set $\{1, \ldots, m\}$, resulting in a group of cardinality $|S_m| = m!$. In our experimental setup, each permutation $\pi \in S_m$ is bijectively mapped to a unique integer token in $\{0, 1, \ldots, m! - 1\}$. Given an input sequence of $n$ permutations $\mathbf{x} = (\pi_1, \pi_2, \ldots, \pi_n)$, the model is required to compute their sequential composition:

$$y = \pi_n \circ \pi_{n-1} \circ \cdots \circ \pi_1, \tag{3}$$

where $\circ$ denotes the permutation composition operator. This task significantly elevates the complexity of state tracking, as the model can no longer rely on the order-invariance property characteristic of abelian groups.

**Algebraic Significance:**   The symmetric group $S_m$ serves as the canonical non-commutative structure for evaluating state tracking. Its fundamental importance is grounded in *Cayley's Theorem*, which states that every finite group $G$ is isomorphic to a subgroup of the symmetric group $S_{|G|}$ Dummit et al. (2004). Hence, by analyzing performance on $S_m$, we effectively probe the model's capacity to internalize the transition dynamics of any finite discrete group.

As noted in Figure 7, we observe the same patterns described in Section 3 and Figure 2, supporting the generalization of these findings and the subsequent arguments to non-commutative state-tracking tasks.

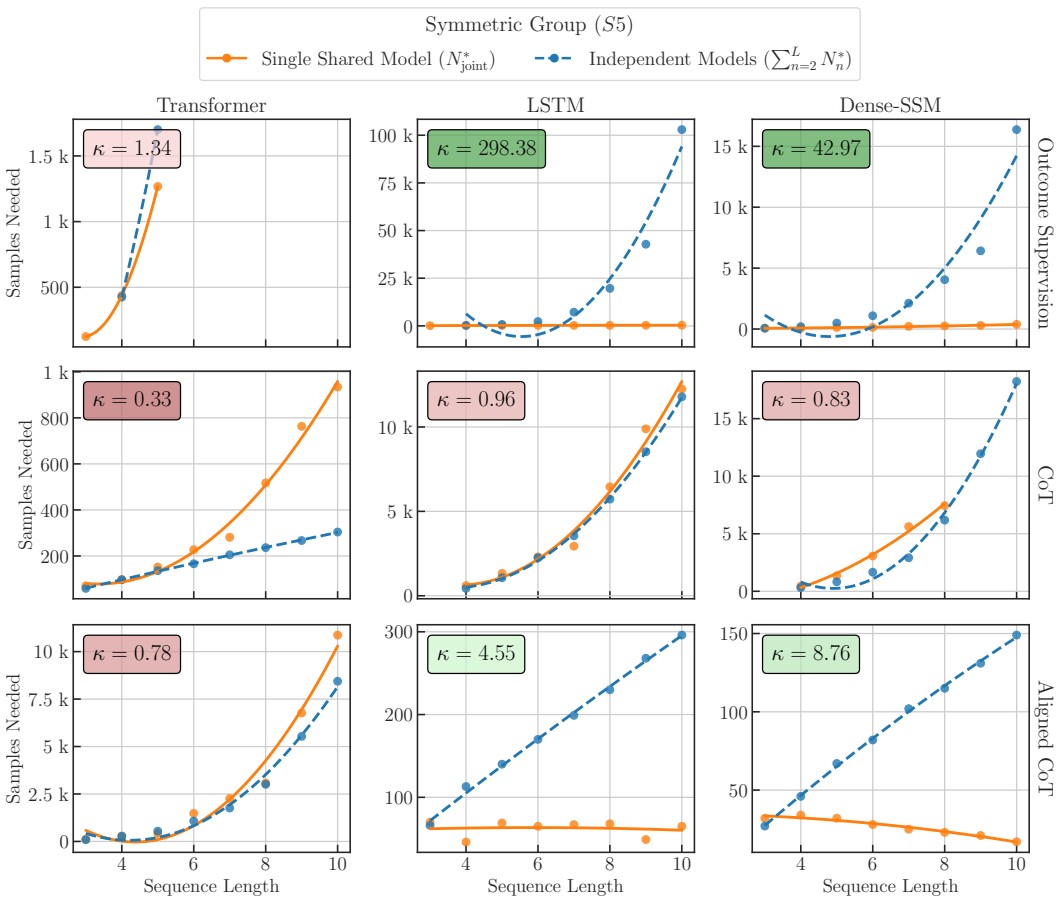

Figure 7: Similar to Figure 2, but for permutation composition task. The results suggest that transformers learn largely isolated solutions for each sequence length.

## C.4  DATA EFFICIENCY EVALUATION FOR SMALLER MODELS

Table 3: $N^*$ for LSTMs with 256 and 768 hidden dimensions. We observe similar trends across both model sizes.

| Model | Format | Mod. | Fixed | | | | Uniform | | | | Short-to-Long | | | |
|---|---|---|---|---|---|---|---|---|---|---|---|---|---|---|
| | | | 5 | 10 | 20 | 30 | 5 | 10 | 20 | 30 | 5 | 10 | 20 | 30 |
| | Outcome Supervision | 2 | 14 | 202 | 257 | 413 | 26 | 34 | 67 | 109 | 12 | 13 | 26 | 16 |
| | | 3 | 151 | 620 | – | – | 84 | 122 | 317 | 686 | 81 | 180 | 132 | 178 |
| | | 5 | 1.3K | 6.7K | – | – | 487 | 876 | – | – | 788 | 1.4K | – | – |
| | | 10 | 6.3K | – | – | – | 3.4K | – | – | – | 5.1K | – | – | – |
| | | 15 | 21.9K | – | – | – | 11.1K | – | – | – | – | – | – | – |
| | | 20 | – | – | – | – | – | – | – | – | – | – | – | – |
| | | 50 | – | – | – | – | – | – | – | – | – | – | – | – |
| | | 75 | – | – | – | – | – | – | – | – | – | – | – | – |
| | | 100 | – | – | – | – | – | – | – | – | – | – | – | – |
| LSTM Dim = 256 | CoT | 2 | 26 | 346 | 3.2K | 14.9K | – | 853 | 24.6K | 306.4K | – | 920 | 581.1K | – |
| | | 3 | – | 5K | 20.5K | – | – | 12.2K | 129.2K | – | – | 29.9K | – | – |
| | | 5 | – | 9.6K | 32K | – | – | 26.3K | 68.1K | – | – | 2.5M | – | – |
| | | 10 | 10.4K | 14.8K | 15.4M | – | 21.7K | 53.1K | – | – | 21.6K | – | – | – |
| | | 15 | 19.3K | 21.1K | – | – | 46.3K | 162.1K | – | – | 73.7K | – | – | – |
| | | 20 | 30.9K | 21.6K | – | – | 64.6K | 351.8K | – | – | 276.7K | – | – | – |
| | | 50 | 33.7K | 36.5K | – | – | 413.8K | 14.7M | – | – | 8.4M | – | – | – |
| | | 75 | 41.1K | 50.5K | – | – | 595.1K | – | – | – | – | – | – | – |
| | | 100 | 46.4K | 13.4M | – | – | 3.8M | – | – | – | – | – | – | – |
| | Aligned CoT | 2 | 8 | 8 | 10 | 8 | 4 | 8 | 2 | 8 | 12 | 7 | 10 | 8 |
| | | 3 | 10 | 11 | 8 | 8 | 21 | 17 | 8 | 8 | 27 | 32 | 39 | 39 |
| | | 5 | 95 | 75 | 52 | 39 | 144 | 101 | 70 | 54 | 142 | 148 | 225 | 209 |
| | | 10 | 324 | 227 | 205 | 209 | 477 | 379 | 317 | 257 | 606 | 1.2K | 1.7K | 1.9K |
| | | 15 | 543 | 502 | 452 | 467 | 846 | 677 | 637 | 772 | 1.4K | 4K | 8.4K | 57.6K |
| | | 20 | 993 | 902 | 919 | 1K | 1.2K | 1.3K | 1.2K | 1.2K | 2.6K | 10.1K | 176.3K | 194.9K |
| | | 50 | 4.6K | 5.3K | 7.3K | 4K | 5.8K | 7.3K | 8.7K | 11.5K | 20K | 168.2K | – | – |
| | | 75 | 9.2K | 9.2K | 13.5K | 15.4K | 10.3K | 12.2K | 15.7K | 20K | 42.6K | 1.2M | – | – |
| | | 100 | 14.5K | 15.6K | 19.8K | 32.6K | 16K | 23.9K | 25.3K | 9.1M | 78.6K | – | – | – |
| | Outcome Supervision | 2 | 14 | 101 | 252 | 309 | 16 | 21 | 64 | 54 | 12 | 9 | 13 | 16 |
| | | 3 | 195 | 620 | 1.7K | – | 78 | 122 | 317 | 526 | 83 | 90 | 90 | 100 |
| | | 5 | 2K | 5.9K | – | – | 387 | 707 | – | – | 381 | 22.7K | – | – |
| | | 10 | 6.4K | – | – | – | 2.2K | 8.6K | – | – | 2.2K | – | – | – |
| | | 15 | 18.8K | – | – | – | 11.8K | – | – | – | – | – | – | – |
| | | 20 | – | – | – | – | – | – | – | – | – | – | – | – |
| | | 50 | – | – | – | – | – | – | – | – | – | – | – | – |
| | | 75 | – | – | – | – | – | – | – | – | – | – | – | – |
| | | 100 | – | – | – | – | – | – | – | – | – | – | – | – |
| LSTM Dim = 768 | CoT | 2 | 20 | 307 | 4.7K | 8.6K | 45 | 692 | 13.4K | 37.1K | 48 | 843 | 1.1M | – |
| | | 3 | – | 2.9K | 9.3K | 14K | – | 7.4K | 40.3K | 5.2M | – | 29K | – | – |
| | | 5 | 1.6K | 10.3K | 20.4K | 44K | 2K | 28.4K | 68.1K | 2.1M | 2.1K | – | – | – |
| | | 10 | 6.1K | 15.7K | 29.1K | – | 14.3K | 544.4K | 500K | – | 16.8K | – | – | – |
| | | 15 | 15.8K | 21.1K | 42K | – | 32K | 67K | – | – | 65.6K | – | – | – |
| | | 20 | 21.2K | 29.6K | 54.3K | – | 40.7K | 409.1K | 14M | – | 215.4K | – | – | – |
| | | 50 | 43K | 50.4K | – | – | 206K | 2.2M | – | – | 10.4M | – | – | – |
| | | 75 | 56.5K | 58K | – | – | 4M | – | – | – | – | – | – | – |
| | | 100 | 84.2K | 9.5M | – | – | – | – | – | – | – | – | – | – |
| | Aligned CoT | 2 | 4 | 6 | 2 | 2 | 4 | 4 | 2 | 2 | 9 | 7 | 7 | 8 |
| | | 3 | 10 | 8 | 8 | 6 | 18 | 12 | 6 | 8 | 27 | 28 | 34 | 34 |
| | | 5 | 95 | 70 | 39 | 31 | 153 | 107 | 80 | 47 | 142 | 148 | 253 | 235 |
| | | 10 | 278 | 194 | 146 | 130 | 447 | 313 | 247 | 232 | 685 | 1.3K | 1.8K | 2K |
| | | 15 | 534 | 478 | 409 | 371 | 865 | 631 | 567 | 500 | 1.5K | 3.9K | 5.2K | 7.1K |
| | | 20 | 798 | 672 | 652 | 798 | 1.3K | 934 | 870 | 820 | 2.7K | 9.4K | 13.4K | 25.9K |
| | | 50 | 3.7K | 4.6K | 6K | 7.4K | 4.1K | 5.3K | 6.1K | 6.9K | 14.7K | 156.2K | – | – |
| | | 75 | 7.4K | 9.2K | 12.2K | 13.3K | 8.4K | 11.3K | 14.2K | 16.8K | 41.1K | 697.4K | – | – |
| | | 100 | 12.8K | 16.6K | 21.9K | 23.7K | 13.3K | 19.4K | 24.3K | 28.5K | 99.7K | – | – | – |

Table 4: $N^*$ for transformers with 2 and 6 layers. We observe similar trends across both model depths.

| Model | Format | Modulus | Fixed | | | | Uniform | | | | Short-to-Long | | | |
|---|---|---|---|---|---|---|---|---|---|---|---|---|---|---|
| | | | 5 | 10 | 20 | 30 | 5 | 10 | 20 | 30 | 5 | 10 | 20 | 30 |
| Transformer 2 Layers | Outcome Supervision | 2 | – | 594 | – | – | 48 | 807 | – | – | 48 | 971 | – | – |
| | | 3 | 134 | – | – | – | – | – | – | – | – | – | – | – |
| | | 5 | 1.5K | – | – | – | 1.7K | – | – | – | – | – | – | – |
| | | 10 | – | – | – | – | – | – | – | – | – | – | – | – |
| | | 15 | – | – | – | – | – | – | – | – | – | – | – | – |
| | | 20 | – | – | – | – | – | – | – | – | – | – | – | – |
| | | 50 | – | – | – | – | – | – | – | – | – | – | – | – |
| | | 75 | – | – | – | – | – | – | – | – | – | – | – | – |
| | | 100 | – | – | – | – | – | – | – | – | – | – | – | – |
| | CoT | 2 | 14 | 26 | 21 | 23 | 37 | 214 | 724 | 1.1K | 42 | 824 | 1M | – |
| | | 3 | 19 | 24 | 25 | 31 | 93 | 456 | 1.3K | 1.7K | 117 | 28.8K | – | – |
| | | 5 | 32 | 31 | 34 | 39 | 241 | 893 | 1.8K | 2K | 653 | 2.4M | – | – |
| | | 10 | 86 | 61 | 52 | 54 | 600 | 1.8K | 3.6K | 4.1K | 11.1K | – | – | – |
| | | 15 | 160 | 97 | 79 | 62 | 805 | 2.9K | 7.4K | 6.8K | 54.2K | – | – | – |
| | | 20 | 229 | 142 | 97 | 78 | 1.3K | 3K | 8.5K | 10K | 168.8K | – | – | – |
| | | 50 | 795 | 542 | 334 | 272 | 3.2K | 6.2K | – | 16M | 6.5M | – | – | – |
| | | 75 | 1.5K | 958 | 688 | 600 | 5.1K | 15.1K | 70.8K | – | – | – | – | – |
| | | 100 | 2.1K | 1.5K | 1.1K | 1K | 6.8K | 20.3K | 33.6K | 209.2K | – | – | – | – |
| | Aligned CoT | 2 | 26 | 197 | 3K | 2K | 37 | 439 | 5.2K | 363.8K | 36 | 869 | 1.1M | – |
| | | 3 | 129 | 2.1K | 7.5K | – | 239 | 11.1K | – | – | 234 | 29.9K | – | – |
| | | 5 | 1.1K | 6.5K | – | – | 1.6K | 40.9K | – | – | 1.5K | 2.5M | – | – |
| | | 10 | 2.1K | 12.4K | – | – | 4.6K | 156.1K | – | – | 11.9K | – | – | – |
| | | 15 | 3.1K | 31.2K | – | – | 8.2K | 1.6M | – | – | 55.3K | – | – | – |
| | | 20 | 4.2K | 59.5K | – | – | 11.2K | – | – | – | 175.5K | – | – | – |
| | | 50 | 10.1K | – | – | – | 15.8K | – | – | – | 6.8M | – | – | – |
| | | 75 | 21.6K | – | – | – | 33.8K | – | – | – | – | – | – | – |
| | | 100 | 25K | – | – | – | 99.8K | – | – | – | – | – | – | – |
| Transformer 6 Layers | Outcome Supervision | 2 | 19 | 364 | – | – | 37 | 549 | – | – | 45 | 913 | – | – |
| | | 3 | 119 | – | – | – | – | – | – | – | 243 | – | – | – |
| | | 5 | 1.1K | – | – | – | 2.6K | – | – | – | 1.5K | – | – | – |
| | | 10 | – | – | – | – | – | – | – | – | – | – | – | – |
| | | 15 | – | – | – | – | – | – | – | – | – | – | – | – |
| | | 20 | – | – | – | – | – | – | – | – | – | – | – | – |
| | | 50 | – | – | – | – | – | – | – | – | – | – | – | – |
| | | 75 | – | – | – | – | – | – | – | – | – | – | – | – |
| | | 100 | – | – | – | – | – | – | – | – | – | – | – | – |
| | CoT | 2 | 10 | 16 | 19 | 20 | 36 | 198 | 744 | 1K | 39 | 824 | 1M | – |
| | | 3 | 16 | 18 | 21 | 25 | 78 | 465 | 1.2K | 1.4K | 108 | 28.8K | – | – |
| | | 5 | 33 | 30 | 31 | 34 | 148 | 1.1K | 1.7K | 1.7K | 647 | 2.4M | – | – |
| | | 10 | 94 | 66 | 62 | 70 | 427 | 2.3K | 3.4K | 4K | 11K | – | – | – |
| | | 15 | 166 | 116 | 78 | 107 | 709 | 2.5K | 5K | 5.1K | 54.3K | – | – | – |
| | | 20 | 377 | 178 | 116 | 135 | 1K | 3.8K | 5.6K | 7.4K | 168.6K | – | – | – |
| | | 50 | 1.1K | 678 | 553 | 470 | 2.6K | 8.6K | 12K | 11.2K | 6.4M | – | – | – |
| | | 75 | 1.7K | 1.2K | 1K | 946 | 4.2K | 11K | 18.1K | 16.1K | – | – | – | – |
| | | 100 | 2.5K | 1.9K | 1.6K | 1.6K | 5.8K | 13.2K | 21.1K | 19.1K | – | – | – | – |
| | Aligned CoT | 2 | 16 | 121 | 644 | 1.2K | 27 | 313 | 2K | 11.7K | 19 | 856 | 1M | – |
| | | 3 | 91 | 2K | 3.7K | 4K | 180 | 8.3K | 66.8K | – | 207 | 29.7K | – | – |
| | | 5 | 528 | 3.9K | 9.2K | 20.8K | 671 | 14.5K | 2M | – | 909 | 2.4M | – | – |
| | | 10 | 1.3K | 6K | 15.5K | 131.4K | 1.7K | 32.3K | – | – | 11.2K | – | – | – |
| | | 15 | 1.5K | 7.9K | 23.3K | – | 2.4K | 61.5K | – | – | 55K | – | – | – |
| | | 20 | 2.6K | 21.9K | 85.3K | – | 4.5K | 190.9K | – | – | 172.6K | – | – | – |
| | | 50 | 13K | 1M | – | – | 22K | – | – | – | 6.7M | – | – | – |
| | | 75 | 21.1K | 15.2M | – | – | 102.1K | – | – | – | – | – | – | – |
| | | 100 | 23.6K | – | – | – | 8M | – | – | – | – | – | – | – |

