# OpenReview forum: "On the "Induction Bias" in Sequence Models"
_ICLR.cc/2026/Workshop/Sci4DL — Sci4DL 2026_

### Official Review · Reviewer_sptK · 2026-02-06

**Fit:** 3
**Significance:** 3
**Confidence:** 3

**Summary:**

The paper compares the inductive bias of Transformer- and RNN-based sequence models on a simple state-tracking task (modular addition). The authors show that RNNs, with some specific input formats, learn much faster and that it translates to better length generalization. They argue it is because of their "induction bias" towards iterative solutions.

**Strengths:**

- this is a nice paper, that is well written and executed and deserves to be presented at the workshop
- the empirical setup is well designed, makes sense, and enables to see significant differences between architectures

**Suggestions:**

Those are suggestions for extending the work towards a full conference paper:
- the results are mostly descriptive ("Observation ..."). Having some more fine grained explanation than the "induction" bias of RNNs would be valuable. Experiment idea: RNNs without recurrent parameter sharing should learn much slower.
- related to the point above: it is unclear that better sampling efficiency comes from induction bias speeding up learning, or just this bias being more closely aligned to the task considered. It would be great to disembiguate these two hypotheses
- one toy task is not enough to link the observed behavior with state-tracking in general. Having 3-5 such toy tasks would be useful to showcase the robustness of the results.
All of that will help in writing a more comprehensive story.

---

### Official Review · Reviewer_W4zN · 2026-02-15

**Fit:** 3
**Significance:** 3
**Confidence:** 2

**Summary:**

The work conducts an empirical study of the state-tracking ability of three selected sequence models: Transformer Decoder, LSTM, and Dense-SSM. The authors propose using a modulo addition task and compare the effective sample sizes necessary to train these models under various supervision settings. Notably, the work unveils limitations of Transformers by showing that they require a similar or greater amount of data compared to an approach in which a separate model is trained for each length. This contrasts with LSTM and SSM models, which exhibit better data efficiency, signaling stronger in-distribution generalization.

**Strengths:**

1. The paper proposes an interesting hypothesis on the in-distribution abilities of Transformers, uncovering a fundamental limitation of this class of models. Furthermore, it addresses the research gap in in-distribution Transformer analysis; the observations made in the empirical experiments can stimulate future research in this regime.
2. The work is well written and clearly communicates the problem, contributions, and outcomes achieved by the authors, with only minor issues regarding the narrative flow.
3. The methodology and experimental protocol are sound, accurate, and well formulated

**Suggestions:**

- While the paper makes claims about the underlying behavior of the Transformer Decoder (and other models), it analyzes this behavior only through observing final efficacy on the given task. To increase the reliability of the results and avoid drawing conclusions from potentially spurious correlations, it would be desirable to analyze the internal states of the models and empirically show that they indeed leverage different "circuits" at different sequence lengths, or analyze changes to internal representations.
- The introduction lacks definitions of key terms used in the work (line 027: state tracking; line 045: supervision regimes). Readability would improve if even brief definitions were provided.
- Line 056: There is no motivation for the task selection — is modulo addition common in the relevant literature, or does it have special properties that make it desirable for the studies conducted in this paper?
- Line 150: What does it mean that CoT is the most efficient task configuration — does this refer to accuracy?
- Line 161: "previously discussed recall bottleneck" — there is no discussion of the recall bottleneck earlier in the paper.
- Line 215: Context rot was explained, e.g., through the lens of bottlenecks in information flow when an LLM is regarded as a graph. This statement needs more clarification and discussion, especially in light of the related literature.
- Line 446: What are the numbers of trainable parameters for each model? It would also be interesting to see whether the ratio of parameters to dataset size influences the results.
- I would recommend adding a reference to the permutation composition task in the main body of the paper, pointing to the Appendix. This would show that the conclusions hold on a different task, which significantly increases the value of the work.

---

### Official Review · Reviewer_RSTo · 2026-02-26

**Fit:** 2
**Significance:** 2
**Confidence:** 2

**Summary:**

This paper studies Transformers trained on a state tracking task and compares it to recurrent models like LSTM, dense-SSM. They find that Transformers do not learn shared/transferable mechanisms across different lengths for the same task, in contrast to recurrent models.

**Strengths:**

The notion of sharing factor $\kappa$ is very useful in quantitatively understanding mechanism sharing across different lengths for the task. It is also interesting to analyze different task formats like outcome supervision, CoT, aligned CoT etc. on simple toy tasks as the ones discussed here.

**Suggestions:**

Questions / Suggestions

1. I couldn’t completely understand the plots in Fig 2 – for the joint training case, isn’t there a single sample complexity value for all lengths $(=N_{joint}^\star)?$ Also, the plot label for blue curve says “Independent models $(\sum_{n=2}^L N_n^\star)$" but the plot seems to be individual datapoints $N_n^\star$ for each $n \in [2,10]$? Moreover, $\kappa$ in Eq 2 is not length dependent, how is the $\kappa$ indicated in the plots calculated?

2. Recent work on looped Transformers [1] seems very relevant to the research question (induction bias) in this work, and I encourage the authors to study these models in their setup as well.

[1] Saunshi et al., 2025. Reasoning with Latent Thoughts: On the Power of Looped Transformers. Arxiv: 2502.17416

---

### Meta-Review · Area_Chair_cGbs · 2026-03-01

**Recommendation:** Accept

**Metareview:**

Although I do share the concerns of reviewers W4zN and sptK about the certain claims of the paper as the results are obtained by observing final efficacy on the given task. I strongly suggest the authors to update their manuscript in camera ready addressing the concerns. It would be interesting to discuss this work and hence I recommend an accept

---

### Decision · Program_Chairs · 2026-03-02

Accept